# Hypolignification: A Decisive Factor in the Development of Hyperhydricity

**DOI:** 10.3390/plants10122625

**Published:** 2021-11-29

**Authors:** Nurashikin Kemat, Richard G. F. Visser, Frans A. Krens

**Affiliations:** 1Plant Breeding, Wageningen University and Research, P.O. Box 386, 6700 AJ Wageningen, The Netherlands; richard.visser@wur.nl (R.G.F.V.); frans.krens@wur.nl (F.A.K.); 2Department of Agriculture Technology, Faculty of Agriculture, Universiti Putra Malaysia, Serdang 43400, Malaysia

**Keywords:** *Arabidopsis thaliana*, apoplast, hyperhydricity, hypolignification, micropropagation, *p*-coumaric acid, piperonylic acid

## Abstract

One of the characteristics of hyperhydric plants is the reduction of cell wall lignification (hypolignification), but how this is related to the observed abnormalities of hyperhydricity (HH), is still unclear. Lignin is hydrophobic, and we speculate that a reduction in lignin levels leads to more capillary action of the cell wall and consequently to more water in the apoplast. *p*-coumaric acid is the hydroxyl derivative of cinnamic acid and a precursor for lignin and flavonoids in higher plant. In the present study, we examined the role of lignin in the development of HH in *Arabidopsis thaliana* by checking the wild-types (Ler and Col-0) and mutants affected in phenylpropanoid biosynthesis, in the gene coding for cinnamate 4-hydroxylase, C4H (*ref3-1* and *ref3-3*). Exogenously applied *p*-coumaric acid decreased the symptoms of HH in both wild-type and less-lignin mutants. Moreover, the results revealed that exogenously applied *p*-coumaric acid inhibited root growth and increased the total lignin content in both wild-type and less-lignin mutants. These effects appeared to diminish the symptoms of HH and suggest an important role for lignin in HH.

## 1. Introduction

Hyperhydricity (HH), refers to the abnormal growth that is observed in tissues grown in vitro in response to water availability from gelrite media, hormonal imbalance, and tissue culture conditions. This phenomenon causes changes in the plant system at both anatomical and physiological levels such as reduced number of palisade cell layers, irregular stomata, chloroplast degeneration, the presence of a thin cuticle or no cuticle at all, an unorganized spongy mesophyll with large intercellular spaces, long petioles, brittle, and translucent and curled leaves [1,2,3,4,5]. A hypothetical model which describes a sequence of reactions leading to HH and which integrates most of the available biochemical data has been published [6,7,8]. These reactions involving soluble phenols, basic and acidic peroxidases, and the metabolism of auxin and ethylene lead to a deficiency in cellulose and lignin which may explain the anatomical change and hence part of the morphological changes.

In higher plants, the cell wall is one of the first tissues affected by stress signals, which are then transmitted to the cell interior and influence several processes [9]. Lignins are complex cell wall phenolic heteropolymers covalently associated with both polysaccharides and proteins [10]. The deposition of lignins in the cell walls of higher terrestrial plants provides rigidity and structural support to the aerial axis. Lignin fills the spaces in the cell wall between cellulose, hemicellulose, and pectin components, especially in vascular and support tissues like xylem tracheids, vessel elements, and sclereid cells. The polysaccharide components of plant cell walls are highly hydrophilic and thus permeable to water, whereas lignin is more hydrophobic. Beyond the structural function, lignin plays several other important biological roles in plants. Because it is much less hydrophilic than cellulose and hemicellulose, it prevents the absorption of water by these polysaccharides in plant cell walls and allows the efficient transport of water in the vascular tissues. The crosslinking of polysaccharides by lignin is an obstacle for water absorption into the cell wall, thus allows for the efficient translocation of water and solutes over long distances within the vascular systems [11,12].

Lignin is a polymer formed by oxidative coupling of *p*-hydroycinnamyl alcohol monomers (monolignols), which are products of the phenylpropanoid pathway [13]. The phenylpropanoid pathway was key to this evolutionary transition, in that flavonoids and other phenylpropanoid derivatives provide protection from UV-induced DNA damage, and lignin provides structural support to both individual tracheary elements and the stem as a whole [14]. One of the major phenylpropanoid pathway end-products is lignin, a heteropolymer important for the mechanical strength and hydrophobicity of the plant secondary cell wall [15]. The first three biosynthetic reactions in phenylpropanoid metabolism are often referred to as the general phenylpropanoid pathway, because they produce *p*-coumaroyl CoA, a major branch-point metabolite between the production of the flavonoids and the pathway that produces monolignols, lignans and hydroxy-cinnamate conjugates [4,5] (Figure 1). The first step in this pathway is the deamination of phenylalanine by phenylalanine ammonia-lyase (PAL) to produce cinnamate, which is the substrate for the cinnamate 4-hydroxylase (C4H), that generates *p*-coumarate. The next step is hydroxylation at the 3-position of *p*-coumarate by *p*-coumarate 3-hydroxylase (C3H) to produce, in succession, caffeate, ferulate, 5-hydroxyferulate, and sinapate. By the sequential action of 4-coumarate: CoA ligase (4CL), cinnamoyl-CoA reductase (CCR), and cinnamyl alcohol dehydrogenase (CAD), these metabolites are converted into the corresponding monolignols. In the last step of the pathway, cell wall-bound peroxidase (POD) catalyzes the oxidative polymerization of the three *p*-hydroxycinnamyl alcohols (*p*-coumaryl, coniferyl and sinapyl alcohols) that give rise to the *p*- hydroxyphenyl (H), guaiacyl (G), and syringyl (S) units of the lignin polymer, respectively [16,17].

The gene encoding cinnamate 4-hydroxylase (C4H) was one of the first phenylpropanoid pathway genes to be cloned [6,7]. There is one copy of C4H in *Arabidopsis* [18]; thus, any carbon flux through phenylpropanoid metabolism is mediated by the activity of the protein encoded by this single gene. Expression of C4H in *Arabidopsis* is apparent in seedlings soon after seed imbibition, and is observable in most organs during all stages of growth. This expression pattern was corroborated by microarray analysis [19]. The hydroxycinnamate ester sinapoylmalate plays an important role in UV protection in *Arabidopsis thaliana* [20]. Besides, the production of sinapoylmalate was localized in the leaf adaxial epidermis [21]. Sinapoylmalate serves as an excellent genetic marker for mutations in genes that are involved in phenylpropanoid metabolism because of its blue-green fluorescence under UV light [22]. Mutations that decrease the accumulation of sinapoylmalate lead to a reduced epidermal fluorescence (*ref*) phenotype that results from the red fluorescence of chlorophyll in the underlying mesophyll. The *ref* mutants derived from this screen have been instrumental in the isolation of a number of genes involved in phenylpropanoid metabolism [14,23,24].

*Arabidopsis ref3* mutants are characterized in that they constitute an allelic series bearing missense mutations in the gene encoding C4H, which failed to accumulate wild-type levels of sinapoylmalate [14,23]. The map positions for the *REF3* gene were all consistent with the hypothesis that the *ref3* mutation is within the gene encoding C4H. Molecular evidence showed that each mutant allele contained a single G to A transition, which results in a mis-sense mutation: GCG to ACG for *ref3-1* (A306T), AGA for AAA for *ref3-2* (R249K), and GGA to GAA for *ref3-3* (G99E). The missense mutations found in the *ref3* alleles were associated with an array of metabolic changes. In addition to having low levels of sinapoylmalate, leaves of the *ref3* mutants accumulated at least two cinnamate esters that were not found in the leaves of wild-type plants [25]. Levels of condensed tannins in *ref3* seeds were also reduced, and lignin deposition in each of the *ref3* mutants was lower as well. The latter phenomenon appeared to be caused by a decrease in guaicyl (G) lignin monomers, leading to a syringyl/guaiacyl (S/G) ratio higher than that of the wild-type [14]. They found that the decrease was exclusively derived from a reduction in guaiacyl-derived subunit content leading to a substantial increase in the mole percentage of syringyl-subunits in the lignin of the mutant. These metabolic changes are likely due to the altered stability and substrate binding of the mutant C4H proteins which all lead to amino acid substitutions in structural motifs that are highly conserved in the C4H proteins of land plants. The reduction in lignin content in plants carrying the mutated alleles resulted in a collapsed xylem phenotype like those previously observed in other *Arabidopsis* mutants affected in lignin and cellulose biosynthesis [26,27].

*p*-coumaric acid is a hydroxyl derivative of cinnamic acid and a precursor for lignin and flavonoids in higher plant. Addition of hydroxycinnamic acid increases lignin production and inhibits shoot and root growth in pea [28], cucumber [29], canola [30], and soybean [31,32,33]. Piperonylic acid (PIP) is a quasi-irreversible inhibitor of the C4H enzyme (Figure 1). Exogenously applied piperonylic acid (PIP) an inhibitor of C4H, gave a significant effect on lignin synthesis and lignin content in soybean [34]. Moreover, the effect of hydroxycinnamic acid in inhibiting root growth and increasing lignin has been described [16,35,36].

The objective of this study was to investigate the role of lignin on HH development by looking at the relative amounts of apoplastic water and air to the total apoplast volume, root growth, total lignin content, C4H activity, C4H gene expression, and phenylalanine ammonia-lyase (PAL) activity, and at the anatomical structure of leaves in *Arabidopsis thaliana* wild-types (Ler and Col-0) and less-lignin mutants (*ref3-1* and *ref3-3*).

## 2. Results

### 2.1. The Effect of p-Coumaric Acid on Apoplastic Water and Air Volumes in Arabidopsis thaliana Col-0 Seedlings

The physical and chemical state of the medium influence greatly plant development and HH. Manifested mainly in the leaves, HH plants exhibit a higher content of water and lignin deficiency. Plants experiencing HH display translucent, wrinkled, or curled leaves, elongated petioles and look brittle. In comparison with 0.4% (*w*/*v*) gelrite alone (Figure 2B), the symptoms of HH (e.g., curled, glassy appearance, longer petioles) were reduced by adding *p*-coumaric acid to 0.4% (*w*/*v*) gelrite solidified media (Figure 2C). The percentage of the apoplast water volume sharply declined from 59% to 38% and the percentage of the apoplast air volume increased from 42% to 62% by increasing the added amount of *p*-coumaric acid from 10 to 500 µM; the differences were even greater when comparing to 0.4% (*w*/*v*) gelrite (Figure 3). No difference was found between 100 µM and 500 µM *p*-coumaric acid for the volume percentages of apoplastic water and air. This shows that 100 µM was the optimal concentration of *p*-coumaric acid in 0.4% (*w*/*v*) gelrite solidified media in reducing HH.

### 2.2. The Effect of p-Coumaric Acid on Lignin Production and Root Growth Linked to the Development of HH

Phenotypic observations showed that both less-lignin mutant seedlings (*ref3-1* and *ref3-3*) already on Micro-agar medium exhibited the symptoms of HH such as curled, elongated petioles, bigger leaves, and brittle leaves (Figure 4B,F). A similar response was found for *ref3-1* and *ref3-3* seedlings on 0.4% (*w*/*v*) gelrite medium (Figure 4C,G). These results suggested that lignin plays a role in HH. These findings led us to investigate the effect of adding *p*-coumaric acid, the product of the C4H enzyme, to less-lignin mutant seedlings on gelrite medium. Exogenously applied *p*-coumaric acid showed a decrease of the development of HH symptoms in both mutants (Figure 4D,H). Therefore, based on previous results, we determined the percentage of apoplast water and air volumes related to the total volume of the apoplast. The result showed that the percentage of apoplastic water decreased and the percentage of apoplastic air increased by adding the optimal concentration of *p*-coumaric acid at 100 µM to 0.4% (*w*/*v*) gelrite medium (Figure 5).

On top of the effect observed on the percentages of the water and air volumes, we studied further the total lignin content of the seedlings in these treatments. Seedlings exposed to *p*-coumaric acid significantly increased their total lignin levels in both wild-types (Ler and Col-0) as well as in the mutants (*ref3-1* and *ref3-3*) (Table 1). The total amount of lignin in hyperhydric Ler wild-type plants on 0.4% (*w*/*v*) gelrite did not significantly different from *ref3-1* mutant on Micro-agar. The same applies to hyperhydric Col-0 plants where the amount of lignin was also very close to *ref3-3* mutant on Micro-agar. Exogenously applied *p*-coumaric acid in both wild-type (Ler and Col-0) and mutant (*ref3-1* and *ref3-3*) increased the total lignin content about 10% to 22% in the seedlings comparing to seedlings on 0.4% gelrite alone. This demonstrates that lignin plays an important role in reducing HH in *Arabidopsis* seedlings. Figure 6 showed the observations on the root growth of the seedlings on Micro-agar (control), 0.4% gelrite and 0.4% gelrite medium supplemented with 100 µM *p*-coumaric acid. The seedlings treated with *p*-coumaric acid showed a reduction in root growth (adventitious roots) in comparison with 0.4% gelrite alone (Figure 6B,C).

### 2.3. The Effect of Inhibiting Lignin Biosynthesis on HH

Having already ascertained that lignin and HH were affected by the addition of *p*-coumaric acid to the media and hypothesizing that *p*-coumaric acid (the product hydroxylation of cinnamic acid) can be channelled into the phenylpropanoid pathway leading to lignins via the C4H reaction, the wild-type (Ler and Col-0) seedlings were treated with piperonylic acid (PIP) as an inhibitor of the C4H enzyme. The objective was to confirm the role of lignin in HH and by using the inhibitor mimicking the effect and phenotype of the less-lignin mutants which were used in this study. Figure 7B,E showed that PIP added to Micro-agar medium triggered the development of HH on wild-type Ler and Col-0 of 14 days-old seedlings (Figure 7A,D). The seedlings showed the symptoms of HH; curled, elongated petioles, and brittle leaves. The seedlings looked very similar to less-lignin mutant *ref3-1* and *ref3-3* on Micro-agar medium (Figure 7C,F).

Determination of the percentages of apoplastic water and air in the wild-type seedlings treated with PIP showed an increase in the percentage of apoplastic water and a decrease of the percentage of apoplastic air in comparison with the control but the percentage of apoplastic water was still a bit lower in the less-lignin mutant (Figure 8). Because the inhibition of C4H by PIP can have a significant effect on lignin synthesis, subsequent analyses were conducted to determine the changes in total lignin in response to this inhibitor (Table 2). The results indicated that the total lignin in the seedlings treated by PIP significantly differed from those of the controls and that no significant difference was found with the less-lignin mutants (*ref3-1* and *ref3-3*). In order to support the findings of the lignin content determinations, the enzymatic activity of C4H was measured. The *Arabidopsis* wild-type seedlings cultured on PIP treatment showed a significant decrease in C4H activity compared to the control while no significant differences were found with the less-lignin mutant seedlings (Figure 9). This is in agreement with [10] who noted that the *ref3* mutation affected protein stability and enzyme function.

Moreover, there were no significant morphological differences between seedlings treated by PIP or from the less-lignin mutant and those of 0.4% (*w*/*v*) gelrite (data not shown). The results on C4H activity were totally in accordance with the results on total lignin content (Table 2). To further link HH to lignin biosynthesis and in particular to C4H we looked at the expression profile of the lignin biosynthesis-related C4H gene in the wild-type seedlings (Ler and Col-0) and in less-lignin mutant seedlings (*ref3-1* and *ref3-3*) (Figure 10). The expression pattern of the C4Hgene was positively correlated with the total lignin content (rho = 0.821) and the expression level of the C4H gene in the less-lignin mutant was found lower compared to the wild-type.

### 2.4. PAL Activity in Normal and Hyperhydric Arabidopsis thaliana Col-0 Seedlings

PAL is the first enzyme involved in the core and entry pathway of the phenylpropanoid pathway and in the biosynthesis of lignin. In order to confirm the role of lignin in HH, PAL activity of normal and hyperhydric *Arabidopsis thaliana* Col-0 seedlings were measured. PAL activity in hyperhydric leaves was significantly decreased compared to normal leaves (Figure 11) as was the lignin content (Table 1). This confirmed the importance of lignin and lignin biosynthesis in the development of HH.

### 2.5. Leaf Anatomy

To investigate whether any visible cell wall defects could be observed in *ref3-3* seedlings, we looked at the ultrastructure of leaves of the wild-type control with and without PIP treatment and of the mutant *ref3-3* by light microscopy. The palisade cells of wild-type leaves were well-organized (Figure 11); however, the palisade cells in the leaves treated with PIP and the mutant were arranged parallel to the epidermal cells with an altered cell shape (Figure 12B,C). These image results were consistent with the observed curling of PIP treated leaves and mutant leaves. Moreover, wild-type vascular bundles were collateral with lignified vessels (Figure 12D, the arrows point towards the lignified compounds (bluish). On the other hand, differentiation of vascular tissue and celular organization pattern of mutant leaves was observed (Figure 12E) which showed shrunken and poor lignification and hardly any xylem consistent with the low levels of lignin deposited in these leaves (Table 2). The same leaf pattern was applied PIP treated leaves.

## 3. Discussion

The plant shikimate pathway is the entry to the biosynthesis of phenylpropanoids. Anterola and Lewis [37] mentioned that lignin levels are regulated by enzymes together with their corresponding genes C4H, resulting in an increase in H lignin. A decisive step in this important phenylpropanoid biosynthesis pathway is presented by C4H, the second enzymatic step, crucial in lignin synthesis. By blocking this step, the ability of plants to produce lignin is impaired. In addition to a role in support, lignin plays a crucial part in conducting water in the plant due to its polysaccharide components that are hydrophobic compared to other polysaccharide components which are highly hydrophilic. The crosslinking of polysaccharides by lignin is an obstacle for water absorption into the cell wall. As reported by [7], hyperhydric plants have a deficiency in both cellulose and lignin synthesis and [14] identified a single C4H mis-sense mutation in At2g30490 (*ref3*) that severely affected growth and lignin development in leaves of *Arabidopsis*. Combining these two observations led us to investigate the role of lignin in the development of HH.

To confirm a supposed entry of exogenously applied *p*-coumaric acid into the phenylpropanoid pathway, experiments were performed by growing the *Arabidopsis* wild-type (Ler and Col-0) seedlings with different concentration of *p*-coumaric acid. We found that exogenously applied *p*-coumaric acid (100 µM) in both wild-type genotypes was optimal and showed a reduction in HH on 0.4% (*w*/*v*) gelrite medium. Exogenous supply of *p*-coumaric acid (100 µM) to less-lignin mutants (*ref3-1* and *ref3-3*) also resulted in similar effects which a decrease in development of hyperhydric symptoms. Higher concentrations of *p*-coumaric acid might prove to be better in these mutants but this was not tested. Besides, the addition of *p*-coumaric acid to 0.4% (*w*/*v*) gelrite also showed a pronounced decrease in root growth when compared to 0.4% (*w*/*v*) gelrite alone. The inhibitory effect of *p*-coumaric acid treatments on root growth might be related to the increase in lignification, leading to a decrease in root pressure and consequently reducing HH.

Lignification, the metabolic process of sealing a plant cell wall by lignin deposition, occurs during the course of normal tissue development and it is an important step during root growth. The most important finding of the current study was that *p*-coumaric acid not only significantly reduced the root growth of the seedlings on 0.4% (*w*/*v*) gelrite medium but also increased total lignin (Table 1). This finding is of particular interest because the reduction in root growth has been considered one of the first effects of *p*-coumaric acid associated with premature lignification of the cell walls [16,32]. These results were also obtained in parallel in many plant species as documented by [16,33,36,38]. These reports and the recent observations strengthen the notion of a possible influx of *p*-coumaric and ferulic acid into the phenylpropanoid pathway, followed by increases in total lignin that strengthen the cell wall and reduce root growth. Additionally, according to [37] the studies of lignin deposition in tissue cultures have for the most part been oriented towards elucidating the biochemistry of lignin synthesis. Besides, [39,40] showed a similar action of eugenol, cinnamic acid, and its derivatives on the lignification of isolated plant organs.

It is known that *Arabidopsis*
*ref3* mutants, characterized by an allelic series bearing missense mutations in the gene encoding C4H, failed to accumulate wild-type levels of sinapoylmalate [14,41]. The authors demonstrated that the genetic changes in *ref3* mutants had led to altered stability and substrate binding capacity of the mutant C4H enzymes. In our studies, significant decreases in total lignin in the *ref3* seedlings on Micro-agar medium were observed and significant increases in total lignin after the addition of *p*-coumaric acid to the media. These results suggested that exogenously applied *p*-coumaric acid can be channelled into the phenylpropanoid pathway at this metabolic point. Previous reports provided evidence supporting that *ref3* seeds had reduced levels of condensed tannins and were low in lignin deposition [14].

To confirm the role of lignin and the C4H-based phenylpropanoid pathway in HH, additional experiments were conducted to mimic the C4H mutants (*ref3*) by culturing control seedlings with PIP (a specific inhibitor of C4H) on Micro-agar medium. The results showed that the seedlings developed HH symptoms accompanied by significant increases in the percentage of apoplastic water and concurrent decreases in the percentage of apoplastic air. In fact, the seedlings after PIP treatment also demonstrated a reduction in total lignin when compared to the control treatment without PIP. For both apoplastic properties and lignin, it was observed that the seedlings after PIP treatment were not significantly different from the *ref3* mutants. Because PIP inhibits the C4H reaction, thereby blocking the phenylpropanoid pathway, it thus affected the lignin biosynthesis and caused HH. Moreover, transverse sections of the leaves in Figure 12 showed that the C4H modifications caused similar cellular disorganization and cell wall defects in the seedlings treated by PIP and of the less-lignin mutant. Based on our results, we hypothesize that these cellular deficiencies with large intercellular spaces (ICs) and little lignin deposition were responsible for rendering the apoplast more hydrophilic and together with the effect of higher root pressure and high relative humidity in the headspace eventually caused HH. Reduction of root by adding *p*-coumaric acid, decreased the root pressure thus less water absorbed from the gelling medium thereby reduced the symptoms of HH.

The present study demonstrated that the changes in C4H activity affected total lignin. C4H is the first P450-dependent monooxygenase of the phenylpropanoid pathway, and it is widely expressed in various tissues, particularly in roots and cells undergoing lignification [33]. In fact, [42] demonstrated a correlation between C4H activity and the deposition of lignin in phloem and xylem fiber cells of *Zinnia elegans*. Interestingly on top of the changed C4H enzyme activity due to a single amino acid substitution, the C4H gene expression in the *ref* single base pair mutants also was reduced compared to the wild-type. This complied to the findings by [43] who found that the transcript level of RPS4 in the chlorophyll-deficient mutant (*cdm*) of Chinese cabbage, carrying a single base-pair point mutation leading to a single amino acid substitution, was lower than than in the wild-type (Fukuda 50 FT). Together, less transcription and translation, so less enzyme, combined with the reduced activity of the enzyme caused by the mutation-induced amino acid substitutions will have led to a lower lignin.

Besides, our studies demonstrated that the overall anatomical features of leaves of the PIP treated seedlings and of the mutants were significantly altered compared to the wild-type control. The normal leaves (control) contained vascular bundles with lignified xylem elements, whereas hyperhydric ones were poorly lignified on PIP and *ref3-3* mutant. The normal leaves exhibited well-defined palisade formed by single cell layer and layers of spongy mesophyll cells with reduced intercelullar spaces, meanwhile the hyperhydric leaves appeared to have an unorganized spongy mesophyll with large intercellular spaces, the cells were dispered and not as elongated as in the normal leaves. This explained an increase on the percentage of apoplastic water in the apoplast. These findings concur with those of [44] who found cell wall defects occurred on knockout AtCESA2 gene in *Arabidopsis* Ler and the leaves curling much more than the wild-type. Similarly documented by [3,4] the leaves of hyperhydric shoots were characterized by the lack of clear differentiation between the palisade and the spongy parenchyma.

Moreover, PAL is the main enzyme in the initiation of the lignin biosynthesis pathway. A reduction of cell wall lignification may alter the mechanical properties of the cell wall [45]. These changes could lead to a reduction in cell turgor pressure and to a change in water potential, which would further result in the increased water uptake and finally in the occurrence of HH. Our results clearly demonstrated a reversed relationship between PAL activity and the degree of lignification in *Arabidopsis*, confirming the positive effect of lignin on HH. The results were in line [46,47] who also found lower activity of PAL in HH carnation species. The simultaneous development of HH in these experiments suggested that lignin plays an important role in the development of HH.

## 4. Conclusions

The focus of the present work was to investigate the role of lignin in HH. Physiological and anatomical studies, combined with biochemical assays, suggest that lignin plays a pivotal role in the development of HH during in vitro propagation. Our results showed that the possible influx of exogenously applied *p*-coumaric acid, a precursor for lignin, to Gelrite solidified medium, led to increases in total lignin that strengthen the cell wall and reduce root growth. Exogenously applied *p*-coumaric acid reduced HH symptoms, decreased the percentage of the apoplastic water volume, increased the percentage of apoplastic air volume, inhibited root growth and increased the total lignin content in *Arabidopsis thaliana* wild-types and less-lignin mutants plantlets. Further confirmation of the role of lignin and the phenylpropanoid pathway in HH development comes from the results of Arabidopsis *thaliana* less-lignin mutants *ref3-1* and *ref**3-3.* On top of this, exogenous application of PIP, an inhibitor of the C4H enzyme, to *Arabidopsis* wild-type (Col-0) seedlings on Micro-agar medium raised the percentage of apoplastic water and decreased total lignin content. Progress was made in the study of the role of lignin in HH, but more input is required to see how this knowledge can be put to use in preventing HH. Combining *p*-coumaric acid with other beneficial compounds, which could act synergistically in preventing HH, are the next step to be taken.

## 5. Materials and Methods

### 5.1. Plant Materials

*Arabidopsis thaliana* wild-types, Col-0 and Ler, and the less-lignin mutants, *ref3-1* (Ler genetic background) and *ref**3-3* (Col-0 genetic background), seeds were sterilized with 70% (*v*/*v*) ethanol for 1 min and 2% (*w*/*v*) sodium hypochlorite for 15 min and subsequently rinsed three times for 10 min with sterilized distilled water. Then, the seeds were sown in a Petri dish with half-strength Murashige and Skoog (MS) medium including vitamins [48] supplemented with 1.5% (*w*/*v*) sucrose and solidified with 0.7% (*w*/*v*) Micro-agar, pH 5.8. To synchronize germination, the seeds were stratified in the dark for 3 days at 4 °C and after that transferred to a climate room for germination. Growing conditions were at 21 °C with 16 h light/8 h dark (30 µmol m^−2^ s^−1^, Philips TL33). The measurements on apoplastic water, apoplastic air, total lignin, C4H oxidase activity and gene expression analysis were evaluated on 14-days-old seedlings.

### 5.2. Experimental Setup

To investigate the effect of adding *p*-coumaric acid, 7-day-old *Arabidopsis thaliana* wild-type (Col-0) seedlings were transferred to half-strength MS medium including vitamins supplemented with 1.5% (*w*/*v*) sucrose supplemented with *p*-coumaric acid at concentrations of 0, 10, 100, and 500 µM; the media were solidified with 0.7% (*w*/*v*) Micro-agar or 0.4% (*w*/*v*) gelrite. The optimal concentration of *p*-coumaric acid determined from the experiment above was used in the subsequent treatments of the *Arabidopsis thaliana* less-lignin C4H (At2g30490) ethyl methanesulfonate-induced *ref3* mutants, *ref**3-1* from Ler background and *ref3-3* from Col-0 background. To confirm the role of lignin and C4H activity in HH, an experiment with 100 µM PIP in the nutrient medium solidified with 0.7% (*w*/*v*) Micro-agar was conducted on *Arabidopsis thaliana* seedlings of Ler, Col-0 and the less-lignin mutants, *ref3-1* and *ref3.* Half-strength MS medium including vitamins supplemented with 1.5% (*w*/*v*) sucrose solidified with 0.7% (*w*/*v*) Micro-agar acted as control.

### 5.3. Extraction of Cell Walls and Total Lignin Determination

Cell walls were extracted according to [49] with minor modifications. *Arabidopsis* leaves were freeze dried overnight and were ground using mortar and pestle. Each treatment sample consisted of 100 mg powder. The powder was incubated in 10 mL of methanol (MeOH) for 15 min at 20 °C using a thermomixer (Eppendorf^®^ Thermomixer Compact, 5384000020, USA). The suspension was centrifuged at 2750 rpm for 5 min, then the supernatant was discarded. Next, 10 mL of fresh MeOH was added to the pellet and incubated for 30 min at 60 °C, followed by centrifugation at 2750 rpm for 5 min, and again the supernatant was removed. The extraction with MeOH (30 min at 60 °C) was repeated and centrifuged until the supernatant was colourless. The pellet was resuspended in 10 mL of mili-Q water and washed three times. The last pellet was resuspended in 10 mL of 0.5 M phosphate buffer (pH 7.0) containing 5% (*v*/*v*) ethanol (EtOH) and 0.02% (*w*/*v*) protease (Pronase E, Sigma Aldrich). This suspension was incubated for 18 h at 37 °C and centrifuged at 2750 rpm for 5 min, after which resuspension of the pellet in 10 mL of solvent and centrifugation under the same conditions was performed consecutively: 3× with distilled water as solvent, 3× with 95% (*v*/*v*) EtOH, and 2× with absolute EtOH. EtOH residue was removed using vacuum and the final weight of the pellet (cell wall material) was recorded. The total lignin was determined using the acetyl bromide method [50]. Absorption was measured at 280 nm wavelength and optical density inserted in the following ABL equation: X = (Y − 0.0009)/23.077, where X is concentration of lignin (mg/mL), Y is the optical density reading of unknown sample, 0.0009 is the mean intercept value, and 23.077 is the mean extinction coefficient obtained from this work.

### 5.4. Evaluation of Apoplastic Water and Air Volumes in Leaves

The volume of apoplastic water was measured by mild centrifugation [51]. The leaves were excised and weighed, and put in the microfilter centrifuge tube. Leaves were centrifuged at 3000× *g* for 20 min at 4 °C. Immediately after centrifugation, the leaves were reweighed. The apoplastic water volume (Vwater) in μL g^−1^ FW was calculated using the formula [52]: Vwater = [(FW − Wac) × ρH_2_O]/FW. Where FW = fresh weight of leaves in mg, Wac = weight of leaves after centrifugation and ρH_2_O = water density (the water density was taken as equal to 1 g mL^–1^).

The volume of apoplastic air in leaves was measured using a pycnometer with a stopper following the methods of [52]. The leaves were excised, weighed, and then placed into the pycnometer. The pycnometer was then filled with distilled water and the stopper was put in place. The weight of the full pycnometer including leaves, was measured and then the stopper was removed and replaced with a gauze. The pycnometer was placed in a vacuum for 5 min to remove air out of the leaves and this was repeated until all air was removed from the apoplast and the leaves had sunk to the bottom. After that the gauze was removed, the pycnometer refilled with water until it was full without any air-bubbles, it was subsequently dried, and reweighed together with the stopper. The apoplastic air volume (Vair) in μL g^−1^ FW was calculated using the following formula [52]: Vair = [(Wbv − Wav) × ρH_2_O]/FW. Where Wbv = weight in mg of the pycnometer including leaves and water before vacuum infiltration, Wav = weight of the pycnometer including leaves and water after vacuum infiltration, FW = fresh weight of leaves, and ρH_2_O = water density.

The mean percentages of apoplastic water and apoplastic air related to the total apoplast volume were calculated using the following formula %tmvw or %tmva = 100 × tmvw or tmva/Tmap. Here, tmvw was the mean volume of apoplastic water, tmva was the mean volume of apoplastic air, Tmap was the mean total volume of the apoplast, so water + air.

### 5.5. Root Growth Determination

Root growth was recorded by taking a picture from the bottom of petri dishes of seedlings after 14 days of culture on 0.7% Micro-agar (as Control), 0.4% gelrite, and 100 µM *p*-coumaric acid 0.4% gelrite medium. The pictures were used for a visual evaluation and estimation of growth of the roots.

### 5.6. Enzyme Extraction and C4H Activity Assay

C4H was extracted using a modified version of the method of [53]. Fifty milligrams of leaves (fresh weight) were ground to a fine powder in liquid nitrogen using a mortar and pestle. The powder obtained was extracted using a 0.5 mL solution of 0.05 mol/L Tris-HCl buffer (pH 8.9) containing 15 mmol/L β-mercaptoethanol, 4 mmol/L MgCl_2_, 2.5 mmol/L ascorbic acid, 10 mmol/L leupeptin, 1 mmol/L phenylmethylsulfonyl fluoride, and 0.15% (*w*/*v*) polyvinylpyrrolidone and 10% (*v*/*v*) glycerine, and the homogenate was centrifuged at 10,000× *g* for 15 min at 4 °C. The supernatant was collected and then stored at 4 °C for the measurement of C4H activity. For this, the reaction mixtures contained 1.1 mL of 0.05 mol/L Tris-HCl buffer (pH 8.9), consisting of 2 µmol/L trans-cinnamic acid, 2 µmol/L β-nicotinamide adenine di-nucleotide phosphate disodium salt, and 5 µmol/L D-glucose 6-phosphate sodium salt hydrate. Then, 0.15 mL of the extract was added to the mixture to initiate the reaction. The mixture was agitated for 30 min at 25 °C using a thermomixer (Eppendorf^®^ Thermomixer Compact, 5384000020, New York, NY, USA) and the reaction was stopped by adding 50 µL of 6 M HCl. The activity of C4H was assayed by measuring the increase in absorbance at 340 nm using UV- Spectrophotometer. One unit of C4H activity was defined as the amount of enzyme catalyzing an increase in absorbance of 0.01 per minute per miligram fresh weight (total protein).

### 5.7. Quantitative Real-Time PCR (qPCR)

Total RNA was isolated from the leaves (100 mg fresh weight) using an RNeasy^®^ kit (Qiagen, Hilden, Germany) according to the manufacturer’s protocols. The quantity and quality of RNA were determined using NanoDrop 1000™ and gel electrophoresis. The extracted RNA served as template for the synthesis of single-stranded cDNA. cDNA was synthesised using 1 μg of RNA samples with an iScript cDNA Synthesis Kit (Bio-Rad Laboratories, Inc., Hercules, CA, USA). Quantitative Real-Time PCR was performed using the SYBR GREEN super mix (Bio-Rad Laboratories, Inc., Hercules, CA, USA). All qRT-PCR assays were performed as follows: preheating 95 °C for 3 min, 40 cycles of 95 °C for 15 s and 55 °C for 30 s. At the end of the PCR, the temperature was increased gradually from 55 °C to 95 °C to generate the melting curve. The expression of gene C4H: At2g30490 was measured. The expression level of the gene of interest was normalized to the expression level of the reference gene *CYCLIN-DEPENDENT KINASE A1* (CDKA: At3g48750). The relative gene expression was determined based on the 2^−δCt^ calculation method [54]. The primer pairs were retrieved from literature [55]. Primer pair C4H: 5′-CACCGGGAAAGGTCAAGATA-3′ and 5′-CCCAACCTTCACGATTCTGT-3′; CDKA: 5′-ATTGCGTATTGCCACTCTCATAGG-3′ and 5′-TCCTGACAGGGATACCGAATGC-3′.

### 5.8. Determination of Phenylalanine Ammonia-Lyase (PAL) Activity

PAL activity was determined measuring trans-cinnamic acid (CA) content produced as previously described by [56] with slight modifications. Leaf tissue samples weighing 0.3 g derived from 0.7% (*w*/*v*) Micro-agar (Control) and 0.4% (*w*/*v*) gelrite seedlings were homogenized in 0.1M sodium borate (pH 8.8) and centrifuged for 10 min at 4 °C at 12,000× *g*. The supernatant was saved and used for determining protein content using the Bradford procedure. Five hundred microliters of buffer and three hundred and fifty microliters of homogenates as a reaction mixture were pre-incubated at 37 °C for 5 min. The reaction was started by addition of 300 µL 50 mM L-phenylalanine (Sigma–Aldrich, Taufkirchen, Germany). After 3 h of incubation at 37 °C, the reaction was stopped by the addition of 50 µL 5N hydrochloric acid (HCl). All samples were analyzed by spectrophotometry. Parallel controls without L-phenylalanine addition were analyzed to determine plant endogenous trans-cinnamic acid (CA) content. The amount of CA was monitored at 290 nm absorbance.

### 5.9. Microscopy

For lignin deposition, *Arabidopsis thaliana* wild-type and mutant (*ref*) seedlings were fixed in 5% (*v*/*v*) glutaraldehyde solution in 0.1 M phosphate-buffer (pH 7.2) for 2 h at 4 °C. Subsequently, they were washed four times for 15 min in 0.1 M phosphate-buffer (pH 7.2), then twice for 15 min in water. The samples were then dehydrated in a gradient series of ethanol and subsequently embedded in Technovit 7100 (Heraeus-Kulzer Technik, Germany). Infiltration in Technovit was performed according to the manufacturer’s instructions. The samples were then sectioned transversely (5 µm-thick) with a rotary microtome, mounted onto a glass slide, dried, and stained with 0.05% (*w*/*v*) toluidine blue O in phosphate-buffer at pH 6.8. The pictures were made under an Axiophot light microscope (AxioVision software release 4.8.2 (Zeiss, Jena, Germany).

### 5.10. Statistical Data Analysis

For all of the treatments and measurements three repeats of each with 15 plants, were used except for anatomical analysis. The means ± SE are given in the graphs. Data were subjected to one-way analysis of variance (ANOVA) and means were compared using Duncan’s multiple range test at *p* ≤ 0.05. All experiments were carried out at least twice.

## Figures and Tables

**Figure 1 plants-10-02625-f001:**
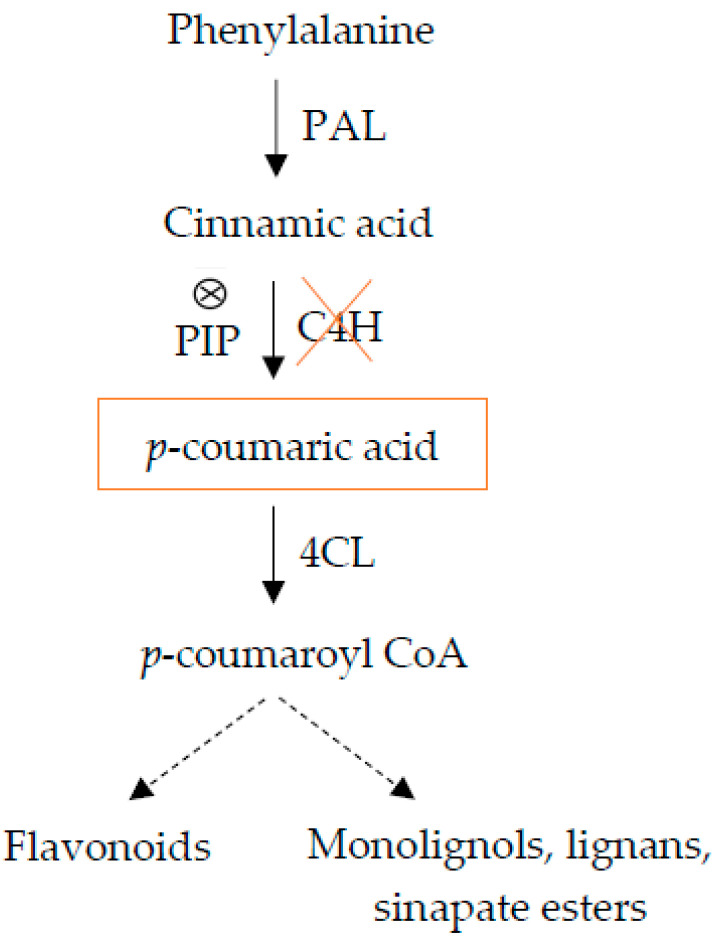
The general phenylpropanoid pathway.

**Figure 2 plants-10-02625-f002:**
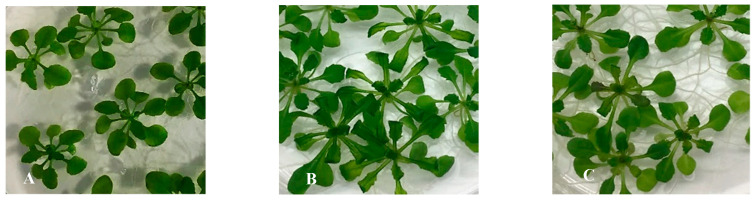
Phenotypic appearance of *Arabidopsis thaliana* Col-0 on *p*-coumaric acid at 14 days of culture. (**A**) Seedlings cultured on 0.7% (*w*/*v*) Micro-agar (control), (**B**) Seedlings cultured on 0.4% (*w*/*v*) gelrite and (**C**) Seedlings cultured on 0.4% (*w*/*v*) gelrite + 100 uM *p*-coumaric acid. Bar = 5 mm.

**Figure 3 plants-10-02625-f003:**
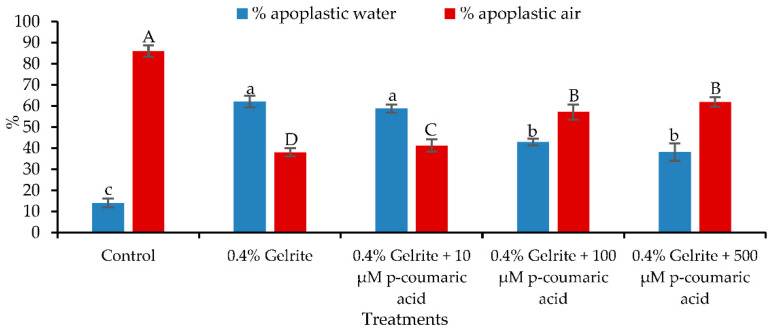
The percentages of apoplastic water and air related to the total volume of the apoplast in *Arabidopsis thaliana* (Col-0) seedlings after 14 days of culture on different concentrations of *p*-coumaric acid. Different letters indicate significant differences between means at α = 0.05 level.

**Figure 4 plants-10-02625-f004:**
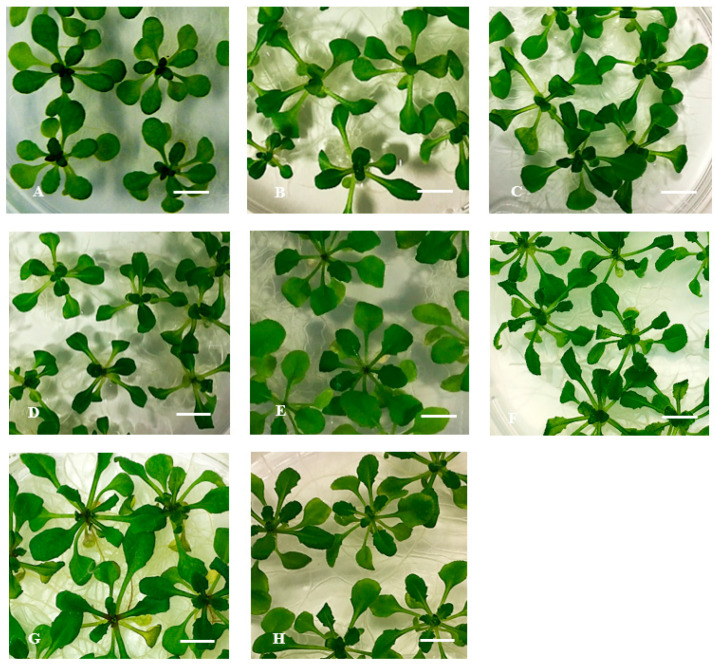
Development of HH on *Arabidopsis thaliana* less-lignin mutants *ref3-1* and *ref3-3* on *p*-coumaric acid after 14 days of culture. (**A**) Ler seedlings cultured on 0.7% Micro-agar (control), (**B**–**D**) *ref3-1*: (**B**) Seedling cultured on 0.7% (*w*/*v*) Micro-agar, (**C**) Seedling cultured on 0.4% (*w*/*v*) gelrite, and (**D**) Seedling cultured on 0.4% (*w*/*v*) gelrite + 100 µM *p*-coumaric acid. (**E**) Col-0 seedlings cultured on 0.7% Micro-agar (control), (**F**–**H**) *ref3-3*: (**F**) Seedling cultured on 0.7% (*w*/*v*) Micro-agar, (**G**) Seedling cultured on 0.4% (*w*/*v*) gelrite, and (**H**) Seedling cultured on 0.4% (*w*/*v*) gelrite + 100 µM *p*-coumaric acid. Bar = 5 mm.

**Figure 5 plants-10-02625-f005:**
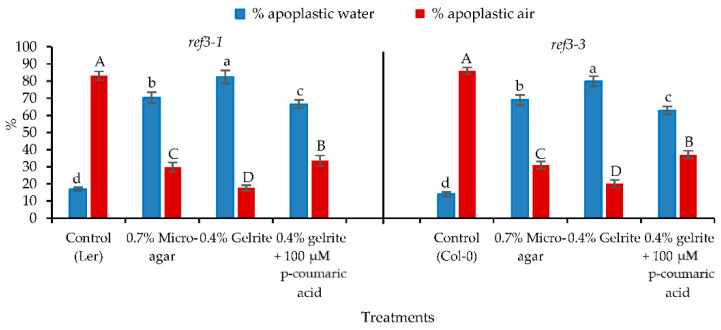
The percentages of apoplastic water and air related to the total volume of the apoplast in *Arabidopsis thaliana* less-lignin-mutant *ref3-1* and *ref3-3* seedlings after 14 days of culture on 0.7% (*w*/*v*) Micro-agar, 0.4% (*w*/*v*) gelrite and 0.4% gelrite (*w*/*v*) + 100 µM *p*-coumaric acid. Different letters indicate significant differences between means at α = 0.05 level.

**Figure 6 plants-10-02625-f006:**
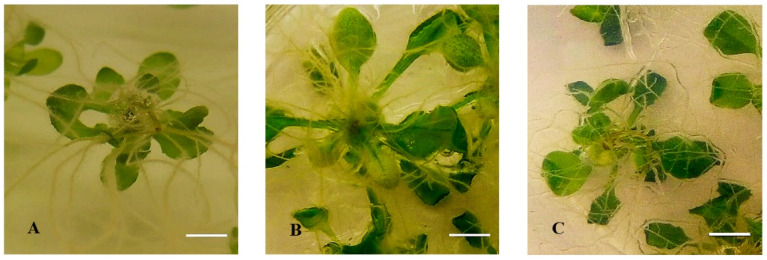
Morphological observations of root growth on *Arabidopsis thaliana* Col-0, and the less-lignin mutant *(ref3-3)*. (**A**) Seedlings on Micro-agar (control), (**B**) Seedlings on 0.4% (*w*/*v*) gelrite, and (**C**) Seedlings cultured on 0.4% (*w*/*v*) gelrite + 100 µM *p*-coumaric acid. Bar = 5 mm.

**Figure 7 plants-10-02625-f007:**
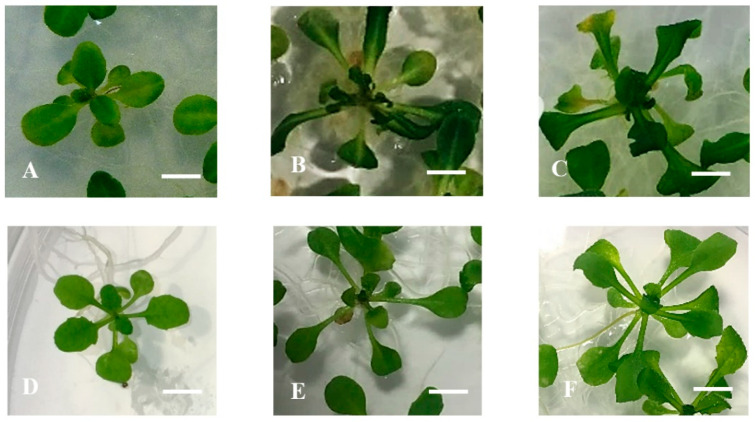
Phenotypic appearance of *Arabidopsis thaliana* Ler, Col-0, *ref3-1,* and *ref3-3* seedlings after PIP treatment on 0.7% (*w*/*v*) Micro-agar (control). (**A**–**C**) Ler seedlings; (**A**) Cultured on Micro-agar (Control), (**B**) Cultured on 100 uM PIP, (**C**) *ref3-1* seedlings, (**D**–**F**) Col-0 seedlings: Cultured on Micro-agar (Control), (**B**) Cultured on 100 uM PIP, and (**C**) *ref3-3* seedlings. Bar = 3 mm.

**Figure 8 plants-10-02625-f008:**
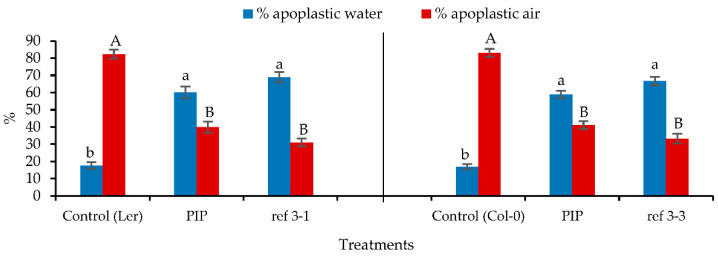
The percentages of apoplastic water and air related to the total volume of the apoplast in *Arabidopsis thaliana* Ler, Col-0, *ref3-1*, and *ref3-3* seedlings on 0.7% (*w*/*v*) Micro-agar after 14 days of culture. Different letters indicate significant differences between means at α = 0.05 level.

**Figure 9 plants-10-02625-f009:**
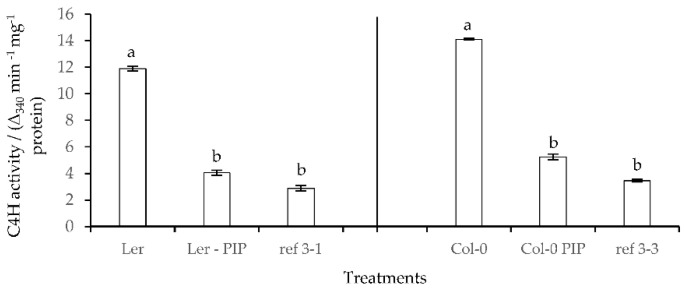
C4H activity of *Arabidopsis thaliana* Ler, Col-0, *ref3-1,* and *ref3-3* seedlings on 0.7% (*w*/*v*) Micro-agar (control) after 14 days of culture. Different letters indicate significant differences between means at α = 0.05 level.

**Figure 10 plants-10-02625-f010:**
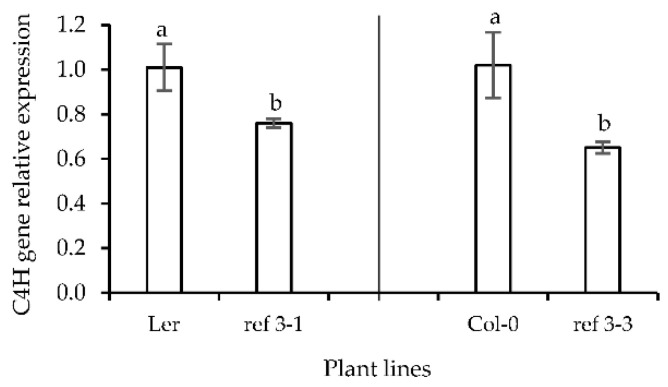
C4H gene expression profile of *Arabidopsis thaliana* wild-type (set at 1.0) and less-lignin mutant seedlings on 0.7% (*w*/*v*) Micro-agar. Different letters indicate significant differences between means at α = 0.05 level.

**Figure 11 plants-10-02625-f011:**
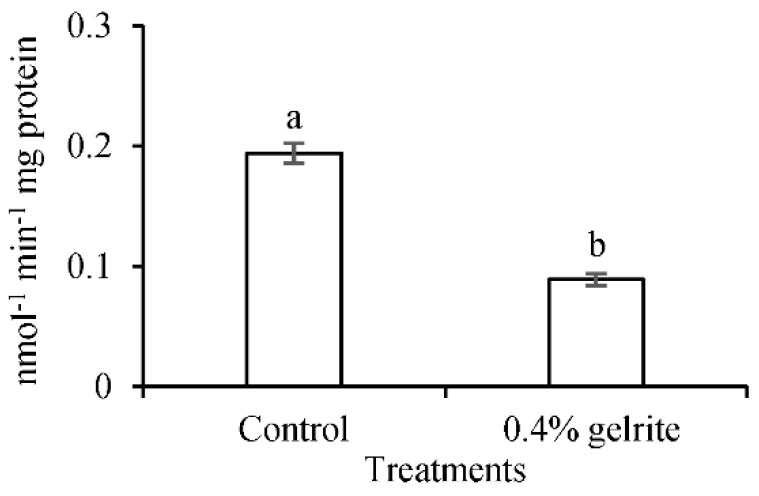
PAL activity of 14 days old *Arabidopsis thaliana* Col-0 leaves on 0.7% (*w*/*v*) Micro-agar (control) and 0.4% (*w*/*v*) gelrite medium. a and b letters indicate significant differences between means at α = 0.05 level.

**Figure 12 plants-10-02625-f012:**
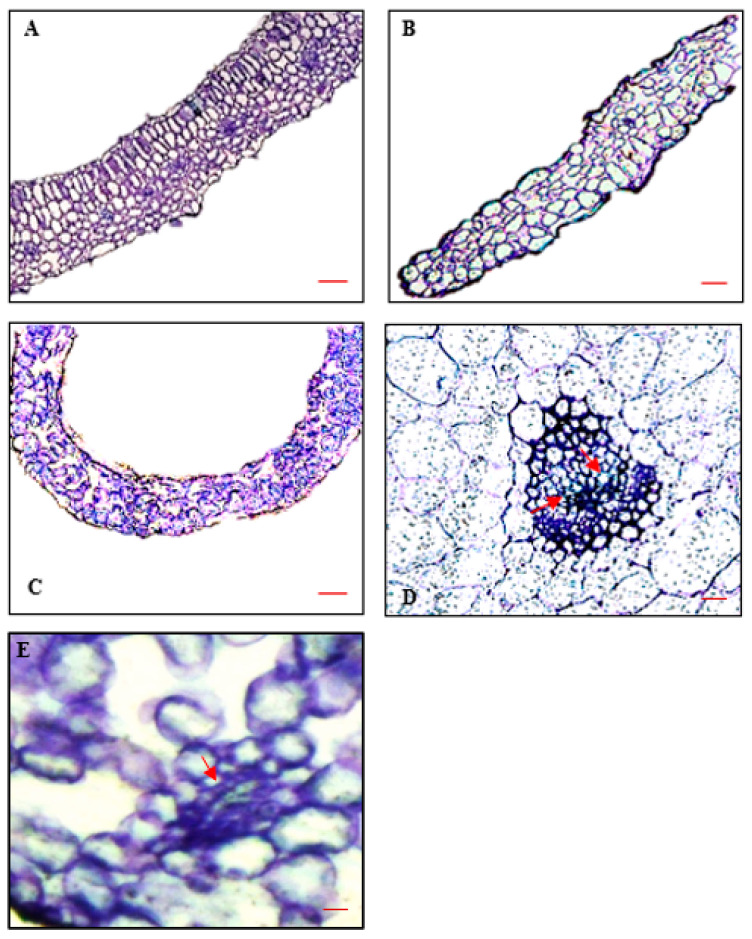
Toluidine blue-stained sections of 14 days old *Arabidopsis* leaves. (**A**) Wild-type (Col-0) on control media, (**B**) Wild-type (Col-0) treated with PIP on Micro-agar, (**C**) *ref3-3* mutant on Micro-agar, (**D**) vascular bundle and cellular organization pattern of normal Wild-type (Col-0) on control media (**E**) hyperhydric *ref3-3* mutant leaves on Micro-agar. Bar = 50 µm (**A**–**C**), 100 µm (**D**,**E**).

**Table 1 plants-10-02625-t001:** Total lignin of *Arabidopsis thaliana* wild-type and less-lignin-mutant seedlings (Ler, Col-0, *ref3-1*, and *ref3-3*).

Line	Lignin (A_280_ mg g^−1^ Cell Walls)
Ler 0.7% Micro-agar (control)	0.0182 ± 0.0002 ^a^
Ler 0.4% Gelrite	0.0084 ± 0.0006 ^d^
Ler 0.4% Gelrite + 100 µM *p*-coumaric acid	0.0123 ± 0.0004 ^b^
*ref3-1* Micro-agar	0.0076 ± 0.0006 ^d,e^
*ref3-1* 0.4% Gelrite	0.0067 ± 0.0005 ^e^
*ref3-1* 0.4% Gelrite + 100 µM *p*-coumaric acid	0.0102 ± 0.0003 ^c^
Col-0 0.7% Micro-agar (control)	0.0225 ± 0.0004 ^a^
Col-0 0.4% Gelrite	0.0119 ± 0.0002 ^c^
Col-0 0.4% Gelrite + 100 µM *p*-coumaric acid	0.0146 ± 0.0007 ^b^
*ref3-3* Micro-agar	0.0098 ± 0.0003 ^d^
*ref3-3* 0.4% Gelrite	0.0088 ± 0.0003 ^d^
*ref 3-3* 0.4% Gelrite + 100 µM *p*-coumaric acid	0.0138 ± 0.0004 ^b^

The means of 9 leaves ± SE are presented; letters indicate significant differences between means at α = 0.05 level.

**Table 2 plants-10-02625-t002:** Total lignin of *Arabidopsis thaliana* wild-type, less-lignin-mutant, and wild-type + PIP on Mico-agar medium.

Line	Lignin (A_280_ mg g^−1^ Cell Walls)
Ler 0.7% Micro-agar (control)	0.0196 ± 0.0004 ^a^
Ler Micro-agar + 100 µM PIP	0.0090 ± 0.0007 ^b^
*ref3-1* Micro-agar	0.0080 ± 0.0003 ^b^
Col-0 0.7% Micro-agar (control)	0.0216 ± 0.0005 ^a^
Col-0 Micro-agar + 100 µM PIP	0.0099 ± 0.0008 ^b^
*ref3-3* Micro-agar	0.0087 ± 0.0004 ^b^

The means of 9 leaves ± SE are presented; letters indicate significant differences between means at α = 0.05 level.

## Data Availability

The data presented in this study are available in this article.

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
