# Peer review of "Hypolignification: A Decisive Factor in the Development of Hyperhydricity"

_plants, 2021, doi:10.3390/plants10122625_

Round 1

Reviewer 1 Report

Nurashikin Kemat et al. described how lignin deficiency is directly linked to hyperhydricity (HH) in the model plant Arabidopsis thaliana cultured in vitro. They demonstrated that WT plants, normally grown on 0.7% microagar, show HH symptoms on Gerlite 0.4% medium, and that the less-lignin mutants ref3-1 and ref3-3 triggers HH-like responses in normal growth condition. They further showed that the ref3 mutants can be partially rescued by treatment of p-coumaric acid (PCA), a precursor of lignin, and that ref3 phenotypes can be mimicked by reducing lignin with treatments of piperonylic acid (PIP), the inhibitor of cinnamate 4-hyeroxlase (C4H), an upstream enzyme of lignin biosynthesis. Finally, the authors showed that the activities of C4H and PAL (another enzyme in the pathway) are reduced upon ref3-induced HH. This manuscript confirmation and refines the connection between HH and lignin content in cultured plants.

The manuscript, despite having some interesting data, has a poorly designed overarching structure, and many contents are either disconnected or are presented in inverted order. This diffuses the main message, and makes the manuscript very hard to read. The authors must improve the clarity in both structure and contents, and also supplement the manuscript with key missing experimental data.

Major comments

The logical connections within and between sections are rather weak. Authors should improve the logics by reordering much of the content and writing, so the message can be more cohesive and comprehensible.

The authors are strongly suggested to rewrite the manuscript, and may consider to follow one of these narratives, or something similar:

Option one

  • Gerlite causes HH (detailed leaf phenotyping and anatomy, apoplastic content). Need for improvement of culturing condition?
  • HH and lignin content is linked.
  • Prove the HH-lignin link: by PCA rescue of WT HH.
  • Prove the HH-lignin link: by reducing lignin in ref3 mutants and PIP treatment (compare detailed leaf phenotype).
  • Prove the HH-lignin link: by (partially) rescuing ref3 mutant with PCA.
  • Investigate mechanism: C4H activity in different conditions and PIP/PAL treatment.

Option two

  • ref3 mutants have HH-like phenotypes: overall organ shapes, detailed leaf anatomy, apoplastic content…
  • compare ref3 mutants with WT HH condition to confirm that indeed ref3 causes HH.
  • Prove the HH-lignin link: by PCA (partial) rescue of WT and ref3 HH.
  • Prove the HH-lignin link: by reducing lignin with PIP and compare to ref3.
  • Prove the HH-lignin link: lignin content measurement in different condition.
  • Investigate mechanism: C4H activity in different conditions and PIP/PAL treatment.

Other major comments

  1. To me it’s not clear that before this investigation (especially in Results 2.2), ref3 mutant phenotypes are directly linked to HH symptoms. Authors should be more precise by saying “HH-like symptoms” etc. and be extra cautious every time when they mention HH and ref3 mutants in the same sentence. Also, to make the revelation stronger, authors must first mention how lignin content connects to HH in WT in different conditions (see also Major point 3).

  1. The rescue of HH-like symptoms of ref3 mutants by PCA is significant but weak, especially since it does not revert the apoplastic water/air content on 0.4% Gelrite. Given that ref3 HH-like symptoms on 0.7% micro-agar are more comparable to WT HH symptoms on Gelrite, and that there is no reversal of apoplastic water/air content for ref3 on 0.4% Gelrite + PCA condition, the authors must test the rescue of ref3 mutants on micro-agar to see if apoplastic water/air content reverts. Also, authors did not test effect of 0.4% Gelrite + 500uM PCA on ref3 mutants. Even if WT PCA effect saturates at 100uM, it does not readily mean that ref3 responses should be too, especially giving their enhanced HH-like symptoms already at control condition. Authors should also perform apoplastic water/air quantifications in this condition.

  1. Line 117-127. Authors should first describe how lignin content is linked to HH, using WT data, then describe effect of PCA and ref3 and the rescue.

  1. Clarity and streamlining: Figure 2, 4, 6, 11 and Table 1, 2, please clearly specify what is control condition. Authors should consider to merge Fig. 1 and 2, 3 and 4, etc. so to be more streamlined, and they won’t need to re-specify “control condition” in legends of even-numbered figures.

  1. What’s the full name of ref3? And what’s it’s connection to C4H? What does PCA supposed to do? All these should appear in introduction…

Minor comments

Line 74. Link between Gelrite and HH symptom was only mentioned once in Introduction (in the second line). It would be nice to remind readers about their link at the beginning of Results, by first describing Gelrite-grown plant phenotypes based on their own analysis (symptom, water content, etc.) before mentioning the effect of PCA. Same of when PAL appearing out of the blue in Results.

The authors should consider to add the metabolic pathway of lignin biosynthesis (simplified) with p-coumaric acid highlighted in introduction next to line 55-59, or in Fig. 1.

Fig 2. “Different letters indicate significant differences between means”. There are upper and lowercase letters. Are they case-sensitive? Pleas specify in legend or change to all upper or lower cases. Also which test was used (should not only be specified in M&M)?

“ref 3” should be “ref3”, no space.

Author Response

Response to Reviewer 1 Comments

Major comment: The logical connections within and between sections are rather weak. Authors should improve the logics by reordering much of the content and writing, so the message can be more cohesive and comprehensible.

Response: Thank you for the comment and suggestions. We do feel the writing structure of this article can be understood by the reader. This is based on the comments of the other 3 reviewers of this manuscript. Considering the suggestion, we add the result of root growth on HH at point 2.2 (Lines 210-221).

The effect of lignin on the development of HH in Arabidopsis can be seen from the results of the study:

2.1: The effect of p-coumaric acid on apoplastic water and air volumes in Arabidopsis thaliana Col-0 seedlings.

- Studies on the optimal effect of PCA reducing HH; 100 µM PCA.

2.2: The effect of p-coumaric acid on lignin production and root growth linked to the development of HH.

- Optimal study of PCA on WT and less lignin mutants (ref), measured total lignin content and seen root growth (observation by naked eye); HH- lignin link.

2.3: The effect of inhibiting lignin biosynthesis on HH.

-Studies involving PIP and investigating the mechanism - C4H reaction.

2.4: PAL activity in normal and hyperhydric Arabidopsis thaliana Col-0 (WT) seedlings.

- Prove, in order to confirm the role of lignin.

2.5: Leaf anatomy.

- Prove to investigate whether any visible cell wall defects could be observed in ref3 seedlings, we looked at the ultrastructure of leaves of the wild type control with and without PIP treatment and of the mutant ref3-3 by light microscopy.

 Point 1 and 3:

  • To me it’s not clear that before this investigation (especially in Results 2.2), ref3 mutant phenotypes are directly linked to HH symptoms. Authors should be more precise by saying “HH-like symptoms” etc. and be extra cautious every time when they mention HH and ref3 mutants in the same sentence. Also, to make the revelation stronger, authors must first mention how lignin content connects to HH in WT in different conditions (see also Major point 3).
  • Line 117-127. Authors should first describe how lignin content is linked to HH, using WT data, then describe effect of PCA and ref3 and the rescue.

Response  1 and 3: Investigation of ref3 mutant phenotypes (less-lignin mutant) for proving that lignin plays a role on HH. A literature review and description of lignin-HH and ref mutants have been included in the introduction (Line 40-144).

Point 2: The rescue of HH-like symptoms of ref3 mutants by PCA is significant but weak, especially since it does not revert the apoplastic water/air content on 0.4% Gelrite. Given that ref3 HH-like symptoms on 0.7% micro-agar are more comparable to WT HH symptoms on Gelrite, and that there is no reversal of apoplastic water/air content for ref3 on 0.4% Gelrite + PCA condition, the authors must test the rescue of ref3 mutants on micro-agar to see if apoplastic water/air content reverts.

Response 2: Yes, we have tested ref3 mutant on micro-agar (0.7% Micro-agar) and measured the percentage of apoplastic water/air (Figure 5).

Also, authors did not test effect of 0.4% Gelrite + 500uM PCA on ref3 mutants. Even if WT PCA effect saturates at 100uM, it does not readily mean that ref3 responses should be too, especially giving their enhanced HH-like symptoms already at control condition. Authors should also perform apoplastic water/air quantifications in this condition.

Response 2: We agree that testing with ref3 on 500µM PCA could enhance the results, however the concentrations didn’t show significant differences with 100µM PCA on WT (results 2.1). So, we only tested ref3 on the optimal concentration. In order to meet the reviewer concern, additional explanation being added (line 168-169 and 200-204).

 Point 4: Clarity and streamlining: Figure 2, 4, 6, 11 and Table 1, 2, please clearly specify what is control condition. Authors should consider to merge Fig. 1 and 2, 3 and 4, etc. so to be more streamlined, and they won’t need to re-specify “control condition” in legends of even-numbered figures.

Response 4: Half-strength MS medium including vitamins supplemented with 1.5% (w/v) sucrose solidified with 0.7% (w/v) Micro-agar acted as control has been mentioned at Material and methods under Experimental setup (Line 574-575). Besides, control condition also being mentioned for figures 2,4,6,11 and Table 1,2.

 Point 5: What’s the full name of ref3? And what’s it’s connection to C4H? What does PCA supposed to do? All these should appear in introduction…

Response 5: Additional detail about the ref3 (Line 113), C4H (Line 101) and the role of PCA in lignin biosynthesis (Line 61-79 and 140-144).

Minor comments:

Line 74. Link between Gelrite and HH symptom was only mentioned once in Introduction (in the second line). It would be nice to remind readers about their link at the beginning of Results, by first describing Gelrite-grown plant phenotypes based on their own analysis (symptom, water content, etc.) before mentioning the effect of PCA. Same of when PAL appearing out of the blue in Results.

Response: Information has been added in the manuscript (Line 157-160 and 360-361).

The authors should consider to add the metabolic pathway of lignin biosynthesis (simplified) with p-coumaric acid highlighted in introduction next to line 55-59, or in Fig. 1.

Response: The simple metabolic pathway is inserted in the manuscript (Line 99).

Fig 2. “Different letters indicate significant differences between means”. There are upper and lowercase letters. Are they case-sensitive? Please specify in legend or change to all upper or lower cases. Also which test was used (should not only be specified in M&M)?

Response: Different letters (upper and lowercase letters) in the figures refer to significant differences between means % apoplastic air (upper case – red) and % apoplastic water (lower case- blue).  I have mentioned the statistical test used in M&M (Line 704-708)

“ref 3” should be “ref3”, no space.

Response: All has been corrected (highlighted in red).

Reviewer 2 Report

In this work the authors describe the relationship between low lignin levels and the previously described hyperhydrocity abnormalities usually affecting in vitro-cultured Arabidopsis plants. This relationship has been previously described by the group, but they give more data that support this feature. However, the writing, the results and the figures obtained must be extensively revised and substantially refined in order to publish the manuscript in Plants.  However I describe some of the main concerns that I have in this manuscript.

Major comments: 

One of my main concerns of the manuscript is that the authors did not showed a quantification of the number of plants and/or phenotypes associated to HH abnormalities and the severity of this abnormalities, more than the % air/water in leaves apoplast. Moreover, the authors solely showed pictures of the plants where, in some cases, the HH defects are not very clear. For example, in Figure 3 HH abnormalities seems to be, visually, more severe in the ref3-3 plants growing in 0.7% microagar (Fig. 3F) than in 0.4% gelrite (Fig. 3G). In the same way, It seems that the severity of the HH abnormalities is higher in ref3-3 plants growth in 0.4%. gelrite+ 100 uM p-coumaric acid (Fig. 3H) than in the plants growth without p-coumaric treatment (Fig. 3G). I think that those are not the best pictures to show those abnormalities and the quality of the pictures in the manuscript should be improved.

In addition, in Figure 3 and 4: one of the main concerns is the lack of analysis of the Ler control plants growing in all the media and the analysis of % of water and air in the apoplast in those plants in the different conditions. As authors demonstrate in Figure 1, Col-0 plants showed changes in HH abnormalities, therefore I wonder what would happen in this Ler lines. I do not know why the authors did not show those results, because they analyzed the lignin quantity in Ler lines under all the treatments, therefore, those analysis must be included in the manuscript.

Moreover, the lignin quantification should be calculated in mg of lignin per g cell wall. To make this, authors should apply the mathematical formula from the paper that authors cite [38]. This calculation is important to have an evaluation how many mg of lignin have a g of cell wall.  

In the same way C4H activity should be quantified as amount of a compound transformed / min-1 mg-1 protein.

Another concern that the manuscript have is the figure 5. The quality of the photographs needs to be improved and, also, the authors did not say hoy many days the seedling had. The bars (= 5mm) are the same in this figure and the others, but in this figure there are only one plant and in the other figures are more plants. In addition, Col-0 plants growth in Microagar usually were smaller than the other plants, and in this figure, the size of the rosette leaves is the same in all cases. In conclusion, I think that the authors should improve this photographs and be more careful with this kind of issues. 

I have also concerns about the visualization of the less growth of adventitious roots in the Figure 9, because the quality of the photographs are so low that I could not asevere those statements.

Minor comments: 

L40-49: this paragraph have several unreferenced statements, references are needed. 

L55-59: "The first three biosynthetic reactions in phenylpropanoid metabolism are often referred to as the general phenylpropanoid pathway, because they produce p-coumaroyl CoA, a major branch-point metabolite between the production of the flavonoids and the pathway 58 that produces monolignols, lignans and hydroxy-cinnamate conjugates", I strongly suggest the authors to describe more these three first reactions, because they are the main focus of the study of this manuscript. 

One of the thing that I think that should be introduced (in introduction section) is the effect of the treatments with p-coumaric acid and their relationship with the lignification process, because is this treatment is key in this manuscript.

L75, L78, L80 and L82: "p-coumaric acid" should be changed to p-coumaric acid.

L80-82 "No difference was found between 100 uM and 500 uM p-coumaric acid for the volume percentages of apoplastic water and air. 100 uM was the optimal concentration of p-coumaric acid". In these sentences uM must be changed for μM 

Figure 2: this figure should be improved. The names of the treatments of the "x" axe should be more separated to avoid missinterpretation. "p-coumaric acid" should be changed to p-coumaric acid

L102-103: "2.2. The effect of p-coumaric acid on lignin production and root growth linked to the development of HH" this title should be changed because the authors did not mention the changes in root growth were not described in this section.

Moreover, Fig. 3 should be improved: The letters (A-H) of each picture should be changed to white color, because sometimes, it is difficult to understand them.

Fig. 4. The x axe names should have p-coumaric acid ("p" in italics and without capital letters) and the names of each treatment should be more separated in order to avoid a misunderstood.

Table 1: The table legend should have a close parenthesis at the final of the sentence. In the statistics, the authors denote "a and b letters indicate significant differences...", however as there are more letters I suggest to change to "letters indicare significant differences...". This should be changed also in table 2.

L117-127: "On top of the effect observed on the percentages of the water and air volumes, we studied further the total lignin content of the seedlings in these treatments. Seedlings exposed to p-coumaric acid significantly increased their total lignin levels in both wild-types (Ler and Col-0) as well as in the mutants (ref 3-1 and ref 3-3) (Table 1). The total amount of lignin in hyperhydric Ler wild-type plants did not significantly differ from that of the ref 3-1 mutant on Micro-agar and in hyperhydric Col-0  it proved very close to the lignin amount of the ref 3-3 mutant on Micro-agar. Total lignin content in the seedlings increased about 10% to 22% after applying p-coumaric  acid in both wild-type (Ler and Col-0) and mutant (ref 3-1 and ref 3-3) comparing to  seedlings on 0.4% gelrite alone, demonstrating that lignin plays an important role in reducing HH in Arabidopsis seedlings." This paragraph needs to be rewritten, because there are repetition of ideas (p-coumaric recover lignin amounts in cell walls of all lines) and the results are poorly described.

Figure 6. Authors should revise the statistics of the % of air in the apopast in the right graph (ref3-3).

Table 2. 100 uM PIP should be changed to μM 

Figure 8. The "x" axe represent plant lines, not treatments.

Figure 9 and 11. These photographs should be improved. In figure 9, I can asevere that there were changes in adventitious roots with those pictures and, although, the HH malformations in leaf section are evident, the photographs have very low quality. In fig. 11.D there are red arrows in the photograph and they were not described in the figure legend and the authors did not denote those arrows in the text. 

Author Response

Response to Reviewer 2 Comments

Major comment: One of my main concerns of the manuscript is that the authors did not showed a quantification of the number of plants and/or phenotypes associated to HH abnormalities and the severity of this abnormalities, more than the % air/water in leaves apoplast. Moreover, the authors solely showed pictures of the plants where, in some cases, the HH defects are not very clear. For example, in Figure 3 HH abnormalities seems to be, visually, more severe in the ref3-3 plants growing in 0.7% microagar (Fig. 3F) than in 0.4% gelrite (Fig. 3G). In the same way, It seems that the severity of the HH abnormalities is higher in ref3-3 plants growth in 0.4%. gelrite+ 100 uM p-coumaric acid (Fig. 3H) than in the plants growth without p-coumaric treatment (Fig. 3G). I think that those are not the best pictures to show those abnormalities and the quality of the pictures in the manuscript should be improved.

Response: Thank you for the comment. We didn’t show the quantification of of the number of plants and/or phenotypes associated to HH abnormalities and the severity of this abnormalities (phenotype appearance) because the quantification are depending heavily on the eye of the beholder which is more subjective to a large degree. Moreover, the extent of HH is variable, some leaves/shoots having severe symptoms and others only slight symptoms, Thus, in our study we focused on the quantitative approach based on the evidences that indicate that water is allocated in the apoplast; by measuring the amount of water and air in the apoplast. Indeed, more severe HH was found on ref3-3 (less-lignin mutant) on 0.4%. gelrite alone (elongated petioles. Yellowish, curled) as compared to 0.4%. gelrite+ 100 uM p-coumaric acid. Same goes to the seedlings on micro-agar media as they were less-lignin mutants.

In addition, in Figure 3 and 4: one of the main concerns is the lack of analysis of the Ler control plants growing in all the media and the analysis of % of water and air in the apoplast in those plants in the different conditions. As authors demonstrate in Figure 1, Col-0 plants showed changes in HH abnormalities, therefore I wonder what would happen in this Ler lines. I do not know why the authors did not show those results, because they analyzed the lignin quantity in Ler lines under all the treatments, therefore, those analysis must be included in the manuscript.

Response: In this manuscript we didn’t test Ler lines in all media treatments due to the symptoms of of HH (% apoplastic water /air) behave same as Col-0 lines. There we no significance different between ecotype baed on our preliminary results (data not shown). Thus in this study we only tested Ler line on control (0.7% micro-agar) media only.  

Moreover, the lignin quantification should be calculated in mg of lignin per g cell wall. To make this, authors should apply the mathematical formula from the paper that authors cite [38]. This calculation is important to have an evaluation how many mg of lignin have a g of cell wall.  

Response: Calculation been inserted in the manuscript (Line 596-599).

In the same way C4H activity should be quantified as amount of a compound transformed / min-1 mg-1 protein.

Response: Has been corrected (Figure 9 and Line 652).

Another concern that the manuscript have is the figure 5. The quality of the photographs needs to be improved and, also, the authors did not say how many days the seedling had. The bars (= 5mm) are the same in this figure and the others, but in this figure there are only one plant and in the other figures are more plants. In addition, Col-0 plants growth in Microagar usually were smaller than the other plants, and in this figure, the size of the rosette leaves is the same in all cases. In conclusion, I think that the authors should improve this photographs and be more careful with this kind of issues. 

Response: Days of seedlings has been added in Line 279.  Photograph Figure 5 (now Figure 7) has been changed and updated.

I have also concerns about the visualization of the less growth of adventitious roots in the Figure 9, because the quality of the photographs are so low that I could not a severe those statements.

Response: Photograph Figure 9 (now Figure 6) has been changed – high resolution.

Minor comments:

L40-49: this paragraph have several unreferenced statements, references are needed. 

L55-59: "The first three biosynthetic reactions in phenylpropanoid metabolism are often referred to as the general phenylpropanoid pathway, because they produce p-coumaroyl CoA, a major branch-point metabolite between the production of the flavonoids and the pathway 58 that produces monolignols, lignans and hydroxy-cinnamate conjugates", I strongly suggest the authors to describe more these three first reactions, because they are the main focus of the study of this manuscript. 

One of the thing that I think that should be introduced (in introduction section) is the effect of the treatments with p-coumaric acid and their relationship with the lignification process, because is this treatment is key in this manuscript.

Response: Information of lignin, phenylpropanoid pathway, ref mutant, p-coumaric have been inserted in the Introduction  (Line 40-45, 52-58 and 67-144).

L75, L78, L80 and L82: "p-coumaric acid" should be changed to p-coumaric acid.

Response: All has been corrected (Line 162, 165,167and 169)

 L80-82 "No difference was found between 100 uM and 500 uM p-coumaric acid for the volume percentages of apoplastic water and air. 100 uM was the optimal concentration of p-coumaric acid". In these sentences uM must be changed for μM 

 Response: All has been corrected (Line 167and 168).

Figure 2: this figure should be improved. The names of the treatments of the "x" axe should be more separated to avoid missinterpretation. "p-coumaric acid" should be changed to p-coumaric acid

 Response: Figure 3 (previous Figure 2) has been corrected.

L102-103: "2.2. The effect of p-coumaric acid on lignin production and root growth linked to the development of HH" this title should be changed because the authors did not mention the changes in root growth were not described in this section.

Response: The root growth result has been inserted under 2.2 The effect of p-coumaric acid on lignin production and root growth linked to the development of HH" (Line 216-220).

Moreover, Fig. 3 should be improved: The letters (A-H) of each picture should be changed to white color, because sometimes, it is difficult to understand them.

Response: Figure 4 (previously Figure 3) has been corrected (Line 239)

Fig. 4. The x axe names should have p-coumaric acid ("p" in italics and without capital letters) and the names of each treatment should be more separated in order to avoid a misunderstood.

Response: Figure 5 (previously Figure 4) has been corrected.

Table 1: The table legend should have a close parenthesis at the final of the sentence. In the statistics, the authors denote "a and b letters indicate significant differences...", however as there are more letters I suggest to change to "letters indicare significant differences...". This should be changed also in table 2.

Response: Descriptions of table 1 and 2 has been corrected (Line 263 and 326).

L117-127: "On top of the effect observed on the percentages of the water and air volumes, we studied further the total lignin content of the seedlings in these treatments. Seedlings exposed to p-coumaric acid significantly increased their total lignin levels in both wild-types (Ler and Col-0) as well as in the mutants (ref 3-1 and ref 3-3) (Table 1). The total amount of lignin in hyperhydric Ler wild-type plants did not significantly differ from that of the ref 3-1 mutant on Micro-agar and in hyperhydric Col-0  it proved very close to the lignin amount of the ref 3-3 mutant on Micro-agar. Total lignin content in the seedlings increased about 10% to 22% after applying p-coumaric  acid in both wild-type (Ler and Col-0) and mutant (ref 3-1 and ref 3-3) comparing to  seedlings on 0.4% gelrite alone, demonstrating that lignin plays an important role in reducing HH in Arabidopsis seedlings." This paragraph needs to be rewritten, because there are repetition of ideas (p-coumaric recover lignin amounts in cell walls of all lines) and the results are poorly described.

Response: The paragraph has been corrected (Line 205-216 ) -  On top of the effect observed on the percentages of the water and air volumes, we studied further the total lignin content of the seedlings in these treatments. Seedlings exposed to p-coumaric acid significantly increased their total lignin levels in both wild-types (Ler and Col-0) as well as in the mutants (ref3-1 and ref3-3) (Table 1). The total amount of lignin in hyperhydric Ler wild-type plants on 0.4% (w/v) gelrite did not significantly different from ref3-1 mutant on Micro-agar. The same applies to hyperhydric  Col-0 plants where the amount of lignin was also very close to ref3-3 mutant on Micro-agar. Exogenously applied p-coumaric acid in both wild-type (Ler and Col-0) and mutant (ref3-1 and ref3-3) increased the total lignin content about 10% to 22% in the seedlings comparing to seedlings on 0.4% gelrite alone. This demonstrates that lignin plays an important role in reducing HH in Arabidopsis seedlings.

Figure 6. Authors should revise the statistics of the % of air in the apopast in the right graph (ref3-3).

Response: The statistics of the % of air in the apopast in the right graph (ref3-3) has been corrected – Figure 8 (Line 310).

Table 2. 100 uM PIP should be changed to μM 

Response: Has been corrected.

Figure 8. The "x" axe represent plant lines, not treatments.

Response: Has been corrected.

Figure 9 and 11. These photographs should be improved. In figure 9, I can a severe that there were changes in adventitious roots with those pictures and, although, the HH malformations in leaf section are evident, the photographs have very low quality. In fig. 11.D there are red arrows in the photograph and they were not described in the figure legend and the authors did not denote those arrows in the text. 

Response: Photographs for Figure 6 (previously 9) has been updated. The arrows in Figure 11 have mentioned in the text (Line 384 – 385).

Reviewer 3 Report

The authors of the present manuscript have done a well-designed study to observe the effect of lignin on the development of HH in Arabidopsis. Apart from few minor issues, the study is comprehensive.

  1. Can authors represent table 1 in form of a graph? Readers have to spend extra time to understand the data as the values go quite low after the decimal.
  2. Authors should add a picture of vascular bundles from the mutants to support their results from section 2.5.
  3. The authors should mention the total RNA amount used for cDNA preparation in the expression studies rather than cDNA concentration.
  4. Line 283, correct figure 12D to 11D.
  5. Abbreviate the species name after the first occurance.

Author Response

Response to Reviewer 3 Comments

Point 1: Can authors represent table 1 in form of a graph? Readers have to spend extra time to understand the data as the values go quite low after the decimal.

Response: Thank you for the comments and suggestion. Nevertheless, we felt that the results of the total lignin content were more easily understood in table than in graph. So here, we remained the result in table form and this also refers to the comments of other 3 reviewers.

Point 2:  Authors should add a picture of vascular bundles from the mutants to support their results from section 2.5.

Response: The picture has added to the manuscript (Figure 12E).

Point 3: The authors should mention the total RNA amount used for cDNA preparation in the expression studies rather than cDNA concentration.

Response: Has been corrected (Line 658 – 661).

Point 4: Line 283, correct figure 12D to 11D.

Response: Has been corrected (Line 376).

Point 5: Abbreviate the species name after the first occurance.

Response: Has been corrected accordingly.

Reviewer 4 Report

In the manuscript entitled “Hypolignification: a decisive factor in the development of hyperhydricity.” the authors analyzed the possible role of lignin in mitigating hyperhydricity in Arabidopsis when growing “in vitro” on gelrite, a common feature observed often in labs when performing in vitro cultures.

The work does is presented in a clearly with consistent results. The discussion is in agreement with the results without overstretching the conclusions.

There are only a few issues on how sentences are sometimes constructed but overall, the manuscript is well written and is clear.

Minor review:

  • Line 121-123 – Grammar is quite confusing. Please rewrite.
  • Lines 296-300 – Please rewrite. It’s confusing. Here, it would be good to refer to the authors instead of just write the number of the reference.

Author Response

Response to Reviewer 4 Comments

Minor review:

Point 1: Line 121-123 – Grammar is quite confusing. Please rewrite.

Response: The sentence has been corrected (Line 208-216).

Point 2: Lines 296-300 – Please rewrite. It’s confusing. Here, it would be good to refer to the authors instead of just write the number of the reference.

Response: The sentence has been corrected (Line 392-395).

Round 2

Reviewer 1 Report

The authors have significantly improved the writing and the inner logic of the manuscript, by adding necessary introduction, connecting sentences, and content reminders. I appreciate this effort. However, the authors did not address my major point on missing experiments:

Because ref3 mutants are already suffering from strong HH-like symptom on 0.7% micro-agar (control condition), their condition and phenotype will only worsen on Gerlite 0.4% which triggers HH-like symptom even in WT. Therefore, the rescue of HH-like symptom of ref3 by 100uM PCA addition on Gerlite 0.4% will at most mildly reduce their HH-like symptom (which is still very strong phenotype), and will not confer a real rescue of the HH-like symptom back to less water, more air content in leaf, like WT on 0.7% micro-agar. This was indeed shown by the authors, indicating that 1. PCA is not a good rescue for lignin synthesis deficiency caused by ref3, 2. PCA is not a good rescue of HH in general.

Therefore, to test the potential epistatic (this should be the prediction based on authors' new Fig. 1) and dose-dependent effect of PCA on ref3/C4H mutant, and to convinced the readers that PCA is in fact a good rescue of lignin-related HH, I called for a "rescue" of ref3 HH-like symptom on 0.7% micro-agar (which already shows strong phenotype) by additional PCA at 100 and 500uM concentrations. By the author's response of having ref3 on 0.7% micro-agar WITHOUT PCA, I believe my phrasing of "rescue" by PCA was not clear enough, and I do apologize. Nevertheless, with authors' intention of rescuing HH with PCA, I imagined that the message should have been obvious.

Back to the experiment, if PCA's effect on HH-like symptom rescue is real, it should also improve ref3 on both Gerlite (author showed but only one concentration and effect very mild) and micro-agar (author didn't show, but may have stronger rescue), and may potentially push leaf air content to be >50%, which will then be a clear and convincing rescue of HH-like symptom. Without this data, the conclusion can only be that PCA mildly reduces leaf water content, but it cannot rescue lignin-related HH, which is against authors' current conclusion. Therefore, authors' must supplement this missing data of ref3 on 0.7 micro-agar + 100 and 500uM PCA, or change their conclusion "the results of the study presented here confirm that lignin plays a 849 pivotal role in the development of HH during in vitro propagation". It does not show pivotal role, and the study does not confirm it.

Same thing for the other missing experiment that I called for: ref3 on 0.4% Gerlite + 500uM PCA. Author argued that +100 or 500uM PCA did not show difference for WT HH phenotype, and concluded that 100uM PCA is saturating for WT plant, and 500uM is no longer needed for other mutants and/or conditions. This is not true. The authors have already proven that ref3 behaves different than WT, 1. by their already strong HH phenotype on micro-agar, 2. by not sufficiently rescued by 100uM PCA, and 3. ref3 supposed to have less natural PCA production, the authors' own hypothesis (see new Fig. 1). Therefore, 100uM PCA most likely will NOT be saturating for ref3, and authors' must provide additional data with 500uM or even higher concentration of PCA on 0.4% Gerlite-grown ref3 mutants, with all phenotypic analyses like growth phenotype, leaf air/water content, and lignin content. Also, I do not endorse authors removing any existing data or discussion on 500uM PCA additions.

I hope with more elaborate explanations and arguments, the authors can realize the significance of the additional experiments for the missing data that I have requested for in the previous round, but were still not provided. The authors must provide these data:

ref3 on 0.7% micro-agar + 100 and 500uM PCA

ref3 on 0.4% Gerlite + 500uM and higher PCA

Without these data, the authors must tone down their conclusion, clearly state that their study only suggests (not confirms) the effect of lignin synthesis in HH to a certain degree that is not pivotal, and acknowledge that adding PCA as lignin synthesis precursor is not sufficient for prevention of HH.

Author Response

Dear Reviewer,

Thank you for the suggestion. Adding experiments on ref3 at 0.7% micro-agar + 100 and 500uM PCA; ref3 at 0.4% Gerlite + 500uM and a higher PCA are certainly interesting. However, higher PCA concentrations on ref3 mutants are not absolutely necessary because we have found the optimum concentration on WT at 0.4% Gerlite + 100uM and saw a clear effect on ref3 mutants. Similarly, for ref3 treatment at 0.7% micro-agar WITHOUT PCA shows an obvious effect compared to WT (control) on HH symptoms. This suggests that lignin plays a role in HH.

We have included in the discussion section of the revised manuscript (page 12, lines 430-458).

We have also included your point as a consideration for future studies in the conclusions. 

Reviewer 2 Report

The authors has been updated all the figures and they included all the text changes that I proposed in the previous report. The manuscript has been improved. 

The only change that I want to ask for is to change the lignin quantifications in the table 1 and table 2. Although authors said that they change the calculation of lignin in the manuscript, in the final version that the authors give us, the calculations of mg of lignin/g of cell walls are missing in the tables. This is important because with the calculations we can have an estimation of the proportion of lignin in cell walls. 

Author Response

Dear reviewer,

Thank you for pointing this out. We agree with this comment. Apologize for missing the unit 'per gram' in table 1 and 2. Therefore, we have inserted the missing unit in table 1 and 2. 

Thank you.

Round 3

Reviewer 1 Report

I appreciate the authors' new, more stringent conclusions, and their acknowledgement of the potential significance of testing higher PCA concentrations in the future. I remain interested in the authors' study.